# Laser-induced heating for the experimental study of critical Casimir forces with optical trapping

Ignacio A. Martínez[1,2]⋆, Artyom Petrosyan[1] and Sergio Ciliberto[1]

**1** University of Lyon, ENS de Lyon, CNRS, Laboratoire de Physique
**2** LCP Group, ELIS Department, Ghent University, Technologiepark 126, 9052 Gent, Belgium

⋆ iamartinez@ucm.es

## Abstract

Critical Casimir interactions represent a perfect example of bath-induced forces at mesoscales. These forces may have a relevant role in the living systems as well as a role in the design of nanomachines fueled by environmental fluctuations. Since the thermal fluctuations are enhanced in the vicinity of a demixing point of a second-order phase transition, we can modulate the magnitude and range of these Casimir-like forces by slight changes in the temperature. Here, we consider two optical trapped colloidal beads inside a binary mixture. The Casimir interaction is controlled by warming the mixture by laser-induced heating, whose local application ensures high reproducibility. Once this two-particle system is warmed, the critical behavior of different observables allows the system to become its self-thermometer. We use this experimental scheme for analyzing the energetics of a critical colloidal system under a non-equilibrium-driven protocol. We quantify how the injected work can be dissipated to the environment as heat or stored as free energy. Indeed, our system allows us to use the fluctuation theorems framework for analyzing the performance of this critically driven toy model. Our work paves the way for future experimental studies on the non-equilibrium features of bath-induced forces and the design of critically driven nanosystems.



# 1 Introduction

Temperature is a physical quantity that defines the amount of energy a system stores. In particular, Brownian motion observed in the mesoscopic systems is an intrinsic feature of the bath temperature. However, controlling temperature in the micrometrical scale in micromanipulation setups is far from standard. The usual approach consists in working with macroscopic thermal baths which modify the temperature of the microscopic system, although light-induced local heating has been applied either directly [1] or by using highly absorbing spots [2]. Nevertheless, the protocols are limited to stationary regimes [3] or to study the thermalization after energy quenches [4].

A non-trivial example of thermal fluctuations with extreme sensitivity to temperature changes is liquid mixture close to its critical demixing point. An upcoming second order phase transition enhances the thermal fluctuations by modifying the correlation length $\xi$ and the relaxation time $\tau$ of the fluctuations-field $\phi$ of the fluid. Once the liquid approaches its critical temperature $T_c$, both parameters of the fluctuations, $\xi$ and $\tau$, diverge from their intrinsic values $\xi_0$ and $\tau_0$ following a universal scaling, $\xi \approx \xi_0 (\Delta T / T_c)^{-\nu}$ and $\tau \approx \tau_0 (\Delta T / T_c)^{-\nu z}$ where $\nu$ and $z$ are the static and dynamic exponents respectively and $\Delta T = T_c - T$ is the distance to the criticality (from now *critical distance*). Critical binary liquid mixtures have been experimentally tested to produce Casimir-like forces between microscopic objects (critical Casimir forces, CCF) [5], to transfer energy in multi-particle systems [6], to react to the bacterial swimming [7] or to generate self-assembly [8], and they have been proposed to induce non-Gaussian fluctuations [9], to react to the chemical affinity of the tracers by changing their viscosity [10] or to play a fundamental role in the intracellular dynamics [11].

Critical Casimir force (CCF) is a paradigmatic case of bath-induced force. In the vicinity of a critical demixing point, the confinement of the thermal fluctuations produces a force between the confining walls. The sense of force depends on the symmetry of the boundary conditions, attractive in the case of symmetric boundaries, and repulsive in the case of antisymmetric ones. In the case of two identically coated spheres of radius $R$ with central positions $x_1$ and $x_2$ acting as confining walls, and under the Derjaguin approximation, Casimir-like potential can be written as:

$$U_{\mathrm{cas}}(d, \xi) = -\frac{AR\pi kT}{\xi} \exp\left(-\frac{d}{\xi}\right), \tag{1}$$

where $d = x_2 - x_1 - 2R$ is the distance between the surfaces, $kT$ is the thermal energy, and $A \approx 1.3$ is a numerical constant from the numerical approximation [12]. Notice how the temperature can be extracted from $U_{\mathrm{cas}}$ since the correlation length $\xi$ is a function of the distance to the criticality.

This article presents a method for accurately adjusting the interaction force between colloids using a laser source. We demonstrate that by gradually modulating the intensity of the laser on the colloidal solution, we can alter the force acting on two beads trapped in optical tweezers and change the energy fluxes. This method holds the potential for understanding the impact of bath-induced forces, which may be crucial in membrane interactions, and for constructing nano-devices that utilize forces tunable by slight temperature changes. Additionally, it opens avenues for quantitative investigations into the non-equilibrium characteristics of critical baths.

# 2 Experimental system

Our critical bath consists of a micelle-solvent mixture, $C_{12}E_5$ in water, at its critical concentration whose demixing point is at $T_c \approx 30.1°C$, see Methods. The sample has an intrinsic
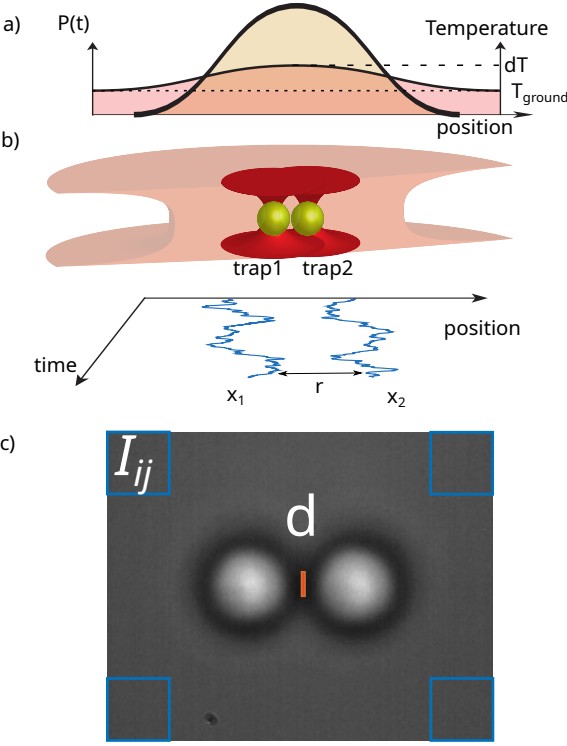

Figure 1: Sketch of the system. a) The heating laser raises the local temperature of an amount ($\delta T$) over the ground temperature ($T_g \approx 28^{\circ}$C). The width of the heating beam ($40\mu$m) is larger than the region of interest (ROI) around the particles (size $\simeq 15\mu$m). The power of the heating laser is slowly changed to access different values of $\delta T$. The temperature field is assumed to be homogeneous and with no thermophoretic flows within the ROI. b) Two particles are held in two independent optical traps (1 and 2). The two particles (2.5 microns radius each) interact via the critical Casimir force produced by confining the concentration field fluctuations between the particle surfaces. c) The two particles are held in two independent tweezers at a distance of $\Delta x_{\text{trap}} = 5570$nm between them. We aim at extracting independent information from the system by using two parameters of each video frame. The positions, $x_1$ and $x_2$, of the particles are tracked while we analyze the dispersion $\delta I^2 = \sum_{i,j}(I_{i,j} - I)^2$ of the intensity at the four corners ($I_{ij}$, 30px x 30 px each) being $I$ the mean intensity).

correlation length of 1.3 nm which is about 5 times larger than previously studied liquids mixtures such as lutidine-water. This feature enhances the critical features, which become non-negligible even at relatively long critical distances. The fluid mixture is contained in a transparent cell of thickness $40\mu$m.

Inside the mixture, two beads (P1 and P2) are held by two optical tweezers (T1 and T2) at a distance of $\Delta x_T$ between their equilibrium positions (see Fig.1). The two beads are equal (Silica, $5\mu$m diameter), so the boundary conditions are symmetric and the critical Casimir force is attractive. Besides the trapping laser beam ($\lambda = 1064nm$) an extra beam of wavelength 1550 nm is sent into the cell (see Fig.1) to modulate the temperature around the particles by the light absorbed by the mixture. This beam has a negligible effect on the trapping strength (see Supplementary Video 1) because it has a focal depth of 400 microns and a diameter of $40\mu$m, much larger than the region of interest (ROI) around the two particles whose size is about $15\mu$m Its power $P_h$ can be changed from 0 to 200mW. Being the cell kept at $T_g \simeq 28.00^{\circ}$C, the critical distance $\Delta T$ in the heated volume is about 2.0 K. The cell is shined by white light to

image the two beads on a CCD camera (see Fig. 1c). The positions $x_1, x_2$ of the two-particle centers are recorded at a sampling frequency of 400Hz at different values of $P_h$.

Furthermore, to obtain an independent measure of the critical distance without including extra devices, we analyze the variance $\delta I^2$ of the illumination light fluctuations in the corners of the image in the spirit of dynamic light scattering Fig. 1c). Since the micelle-rich phase has a different refractive index than the micelle-poor phase, once the transition is crossed we expect a huge change in $\delta I^2$ as a function of $\Delta T$, that allows us to distinguish between the two phases. Furthermore, since the refractive index is expected to depend on the critical distance, we aim at quantifying the behavior of $\delta I^2(\Delta T)$.

## 3 Results

**CCF as nanothermoter.**

In this section, we demonstrate the reproducibility of CCF with light-induced heating and how it can be used as the system's self-thermometer. In the physical system described before, the trap distance $\Delta x_T = 5570$nm is kept fixed while we slightly change the power of the heating laser $P_h(t)$. The following protocol is time-symmetrical and it has a triangular shape with a constant increase of the irradiated power of $v_{\text{heat}} = 90\mu\text{Ws}^{-1} = \max(P_h)/\tau_{\text{heat}}$, where $\max(P_h) = 180$mW is the maximum laser power achieved and $\tau_h = 2000$s is half the duration of the heating cycle, see the black solid line in Fig. 2a) as a guide to the eye. The trajectory of the two particles is followed at a frequency of 400 Hz which is more than enough since the trap stiffness is small ($\kappa = 0.5$pN$\mu$m$^{-1}$) hence the beads relaxation time is about 80 milliseconds.This time is much shorter than $\tau_h$ showing that the heating protocol is quasi-stationary, i.e. the system's temperature is always close to equilibrium. In Fig. 2a), we show the time-evolution of the observables extracted from the analysis of $\delta I^2$ (left axis, blue line) and the beads trajectory (mean distance between the walls $d$ over a 20 seconds block, right axis, red lines). Notice that the total time of the experiment is above one day with the same set of particles. Since the time evolutions of $d$ and $\delta I^2$ are correlated with $P$, we performed an average conditioned to the $P$ values on all of the cycles. The results $\langle d \rangle$ and $\langle \delta I^2 \rangle$ of this conditional average are plotted in Fig. 2b) as a function of $P$. The variance $\langle \delta I^2 \rangle$ has remarkable reproducibility and stability as a function of $P_h$ and shows that the scattering increases when the critical point is approached.

In Fig. 2b), we see that the mean distance $\langle d \rangle$ between the two beads decreases for growing $P_h$ demonstrating the appearance of an attractive force. The appearance of this force can be understood by measuring the probability distribution $\rho(d)$ hence the potential $U(d) \propto \ln(\rho(d))$, which is plotted in Fig. 2c) for two values of $P_h$. At $P_h = 0$ mW, the equilibrium position defined by the Casimir force and the electrostatic repulsion is small compared to that of the optical trap. Instead at $P_h = 100$ mW, i.e. $\Delta T \to 0$ the combination of the different interactions produces an energy potential landscape with two comparable equilibrium positions. For the sake of simplicity during the reading, we will call the optical trap and the Casimir wells 'OW' and 'CW' respectively. Since the profile of the critical force depends on $\Delta T$, the occupation of each well also depends on $\Delta T$ following the detailed balance between the wells. This explains the behavior of $\langle d \rangle$ as a function of $P_h$ in Fig. 2b). The mean value between surfaces $\langle d \rangle$ gets progressively closer from $\langle d \rangle = 500$ nm up to saturation at 110 nm. However, this is not a continuous approach between the particles, but a change in the proportion of the permanence time in OW and CW. Indeed, if we get even closer, the total potential would evolve to a single equilibrium position landscape since the range of the potential scales with $\xi$.

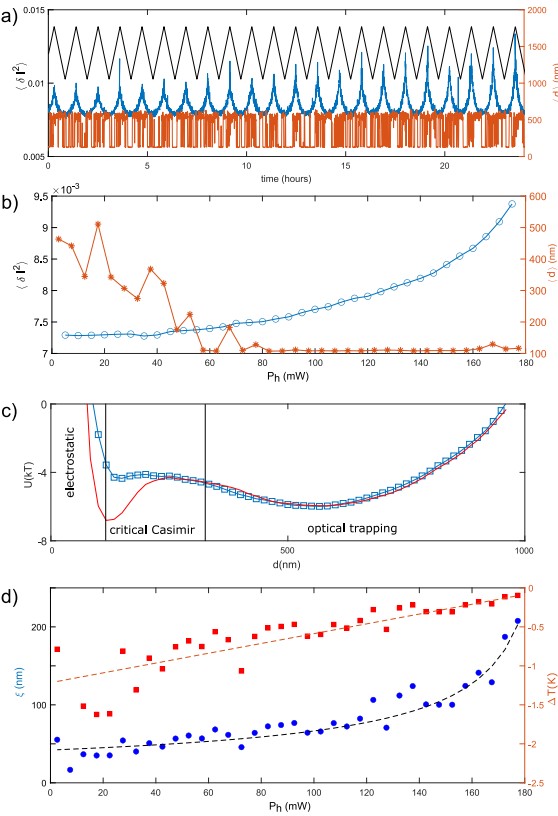

Figure 2: Critical interactions as local nanothermometers. a) Time-evolution of $\delta I^2$ (blue, left axis) and $d = x_2 - x_1 - 2R$ (red, right axis). The black solid line is a guide to the eye without a y-scale showing the power of the heating laser $P(t)$. Notice the reproducibility over tens of repetitions over more than 24 hours of the experiment. b) $\langle \delta I^2 \rangle$ (blues circles and line) and $\langle d \rangle$ (red points) are plotted as a function of the input heating power $P_{\text{heat}}$. For increasing power, the mean distance decreases showing the appearance of an attractive force, whereas $\langle \delta I^2 \rangle$ increases approaching the critical point. c) The attraction potentials measured at two different heating powers $P_h = 0$ mW (blue) and $P_h = 100$ mW (red) are plotted as a function of $d$. The correlation length is extracted from the critical Casimir potential using Derjaguin approximation (blue empty squares), d) The measured $\xi$ is plotted as a function of $P$ (blue dots). Since $\xi/\xi_0 = (\Delta T/T_c)^{-\nu}$, we can infer a critical distance $\Delta T$ for each value of the input power (red points, right axis) obtaining a heating rate of $C_h = (4.87 \pm 0.20)$mKmW$^{-1}$.

As described in previous sections, the correlation length is obtained from $U(d, P_h)$, plotted in Fig. 2c). We fit the critical Casimir force contribution $U_{CC}(d)$ and hence obtain the correlation length $\xi$ of the fluid assuming Derjaguin approximation. Assuming an Ising 3D class of universality, $\xi(\Delta T) = \xi_0(\Delta T/T_c)^{-\nu}$ with $\nu \simeq 0.63$ we can assign a $\Delta T$ to each value of the heating power $P_h$. The measured $\xi$ (blue points) and $\Delta T$ (red dots) are plotted in Fig. 2d) as a function of $P_h$. The blue dashed line is the estimated $\xi(\Delta T)$ with $\nu = 0.63$. This allows us to find a relationship between the temperature $T_{ROI}$ of the ROI and the heating power, specifically $T_{\text{ROI}} = C_h P_h + T_g$ with a heating rate of $C_h \approx (4.87 \pm 0.2)$mK mW$^{-1}$.

Using this relationship between $\Delta T$ and $P_h$, we observe that $\langle \delta I^2 \rangle$ has a power law behavior (blue dashed line in Fig. 2b) as a function of $\Delta T$ but with an exponent much larger of $\nu$. Specifically comparing $\langle \delta I^2 \rangle$ with $\xi$ we find $\langle \delta I^2 \rangle \propto \xi^\alpha \propto \Delta T^{-\nu\alpha}$, where $\alpha = 4.7 \pm 0.6$.

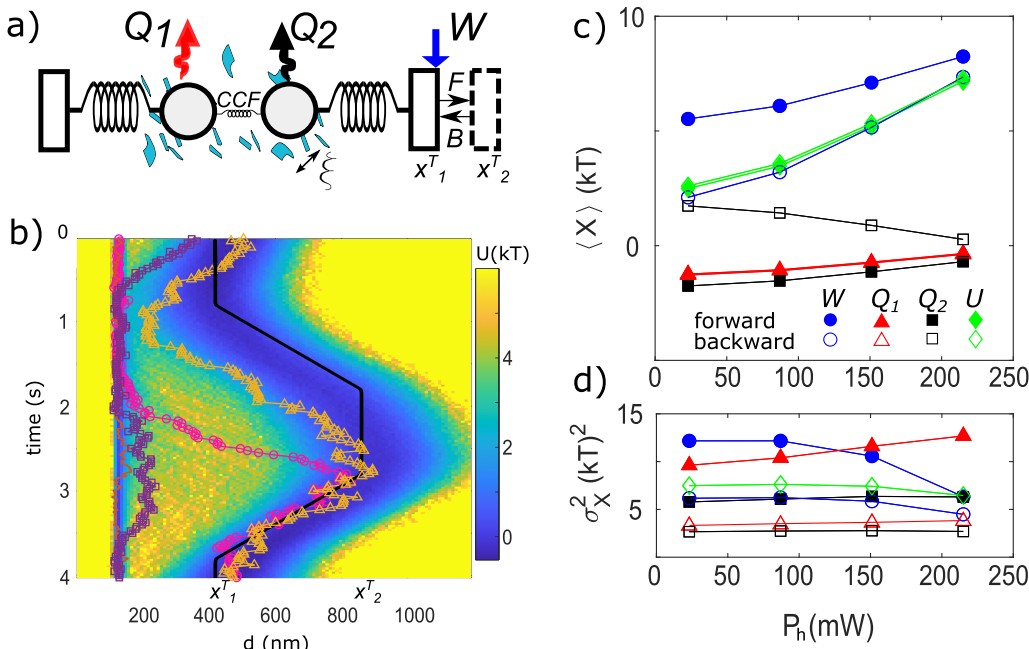

Figure 3: a) Energetic sketch of the system. Two particles are held by independent optical tweezers within a critical bath with correlation length $\xi$ at a given critical distance ($\Delta T$). Trap 1 remains static while trap 2 is moved back and forward at a constant velocity between two equilibrium positions. b) Color plot showing the temporal evolution of the energetic seascape $U(d)$ felt by the relative distance degree of freedom $d = x_2 - x_1 - 2R$. The two-colloidal particle system has different options through the protocol. The single trajectories can start synchronized at the optical trapping well (OW) or the critical well (CW) to remain in the same state, OW $\rightarrow$ OW (yellow triangles) or CW $\rightarrow$ CW (red solid line), or change it, OW $\rightarrow$ CW (purple squares) or CW $\rightarrow$ OW (magenta circles). For the sake of simplicity in the analysis of the energetics, we will focus on the analysis of OW $\rightarrow$ OW and CW $\rightarrow$ CW, defining them as OFF and ON respectively. c-d) Energetics of the system. Work (blue circles), the heat released by particle 1 (red triangles) and heat released by particle 2 (black squared), and the change of internal energy (green diamonds) are shown for the forward and backward process ($X_f$ filled symbols, $-X_b$ empty symbols). c) Mean value $\langle X \rangle$ and d) variance $\sigma_X^2$ as functions of the input energy $P_h$.

## Switching the energy transfer between colloids by light-induced heating

In this section, we apply this laser-induced heating method in a dynamical colloidal system. We designed a toy machine based on a protocol in which thermodynamic magnitudes (work, heat,...) can be easily defined for analyzing their statistics under the stochastic thermodynamics framework. We inject energy as work in P1 for studying how the energy is dissipated via released heat in both particles, see the sketch in Fig. 3a). For that purpose, we displace the position of trap 2 following a time-symmetric protocol $\Gamma$. The position of the movable trap is changed from $\Delta x_1^T = 5500$nm to position $\Delta x_2^T = 5840$nm at a constant velocity $v = 342$nm/s, see black solid line in Fig. 3b). We call the action of pushing T2 away from T1 the forward (F) protocol while the action of getting the traps closer is the backward one (B). Both processes, of one second each, are connected by two equilibrium positions of the same duration to warrant the equilibration of the colloids within their corresponding traps giving a protocol total time of 4 seconds. The stiffness of the traps is kept small to allow a broader exploration of the phase space ($\kappa = 0.5$pN$\mu$m$^{-1}$). The dynamics can be drastically modified by small

temperature changes which determines the ratio between the critical (CW) and optical (OW) wells. Indeed, in Fig. 3b) it is shown how the system, the colloidal particles, can evolve in different ways along the potential seascape represented by the color map in the background image. At the beginning of each cycle, the system can mainly lie either in OW (yellow and purple time-series in Fig. 3b)) or CW (magenta and red). If the traps are close, the thermal transitions between the states are allowed and the system may erase its previous state. Once T2 is pushed away, the system must make a choice, see the different trajectories on Fig. 3b). As the probability of choosing each option depends on the ratio between the well's depth, the performance of the toy machine can be modulated by changing the distance to the criticality hence the heating laser power. In this analysis, our focus will be on two events with higher statistical significance, specifically those that do not alter the well. We refer to the events where particles remain within the optical trap well (OW) as OFF events, while those with the particle residing in the critical well (CW) are labeled as ON events. The trajectories of the ON and OFF events are depicted by red and yellow curves, respectively, in Fig. 3b).

In Fig. 3c) and d) we show how the ensemble average of the system's energetics changes as a function of $P_h$, that is, the temperature. From the trajectories, we obtain the values of the stochastic work ($W$) and both released heats ($Q_1$ and $Q_2$) within the framework of stochastic thermodynamics, see Methods. We calculate the ensemble average of each quantity $\langle X \rangle$ and its probability density function $\rho(X)$. The mean value of the heat released by each particle $\langle Q_i \rangle$ and the mean injected work $\langle W \rangle$ are shown in Fig. 3c) with their standard deviations, Fig. 3d). Indeed, $\langle W \rangle$ increases with temperature during F and B protocol, but there is a discrepancy between them. The same features seem to appear in other magnitudes like the heat released by P2 ($\langle Q_2 \rangle$, black ) but not by P1 (red, $\langle Q_2 \rangle$). Finally, the mean change of internal energy ($\langle U \rangle = \langle W \rangle + \langle Q_1 \rangle + \langle Q_2 \rangle$) remains constant during both protocols as expected although changes with the critical distance. The variance of the same quantities is shown in Fig. 3d), where we observe different behaviors between F and B. Indeed, the existence of different options during the process produces a bimodal distribution in the energetics, see Fig. 4). If we compare the distribution of the injected work $\rho(W)$ for different $\Delta T$, we observe how the critical interactions start to dominate at small $\Delta T$, the CW dominates OW. In Fig. 4 we also compare the distributions of the forward $\rho_F(W)$ and backward $\rho_B(-W)$ protocols in the Crook theorem spirit, $\rho_F(W)/\rho_B(-W) = \exp(W - \Delta F)/kT$. The global energetics of the system is the combination of pure dissipative events (OFF) and events that change the free energy of the system (ON).

The same analysis can be performed for each quantity as it is presented in Fig. 5, which shows how the ON events dissipate much less energy than the OFF events. Indeed, the OFF events are dragging-like processes in a more complex environment (the non-negligible hydro-dynamics due to the surfaces' proximity derives in a non-homogeneous viscosity, the Casimir interactions,...). The ON events allow a higher proximity between the surfaces, and pulling away the traps between them increases the effect of the critical interaction since the increasing distance between the particle and the trap equilibrium position suggests a higher importance of the CCF. It is this increase of the critical Casimir force once the traps are pulled away that increases the free energy of the system while the small change in the relative distance between the particles is the reason for the small energy dissipation during ON events.

## 4 Discussion

Since the temperature of the fluid is increased locally, there is no direct measurement of the temperature via thermometry. Indeed, obtaining a reproducible nano-thermometric technique from our typical experiments is one of the objectives of the article. Here, we have based

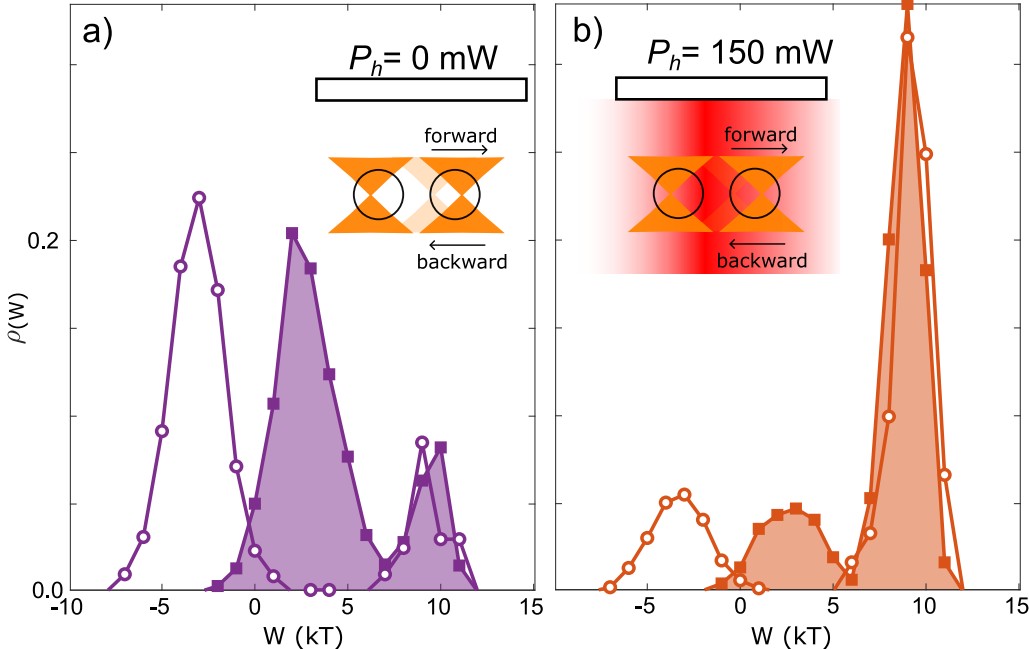

Figure 4: Work injected into the system during the forward $\rho_F(W)$, full squares and colored area, and backward process $\rho_B(-W)$, empty circles. a) Statistics of W far from the criticality, $P = 0$mW with $\xi \approx 65$ nm. b) Close to the criticality, $P = 150$mW with $\xi \approx 140$ nm. The bimodality of all the distributions can be interpreted as the contribution of the two possible equilibrium states. Moreover, the distributions can be decoupled and analyzed in the Crooks-theorem spirit with a crossing of the forward and backward distributions close to zero for the OFF events and a crossing different from zero for the ON events.

our temperature measurement, or more precisely the *critical distance* $\Delta T$ from $T_c$, on i) the trajectory of the particles via CCF, and 2) the fluctuations of the pixels' luminosity. Assuming that the critical temperature is well defined at $T_c = 30.1^{\circ}$C, we can infer the temperature of the fluid as the distance to this reference $T = \Delta T + T_c$. Indeed, this is one of the typical fingerprints of studying CCF using optical traps. The possible local heating of the laser trap and the illumination for video tracking may produce a mismatch between the temperature of the thermal bath and the temperature of the physical system. The influence of the critical distance on the ratio of the two potentials makes the statistics of the jumps as well as the population rates very sensitive to the temperature changes in its environment. However, this method could be used by any physical system to detect temperature changes autonomously, for example, confining a vesicle filled with a critical fluid and sensoring its size or shape.

Using light as the heating source allows us to compare the changes of the different observables, the experiment is clear, very reproducible, and with no dependence on a statistic of jumps, so very accurate. However, the optical configuration of our setup makes difficult an analytical expression for the account of the change of the dispersion with the critical distance via critical exponents, $\langle \delta I^2 \rangle \propto (\Delta T)^{\mu}$, where $\mu = (3.0 \pm 0.5)$ seems to be a combination of different critical exponents. As in previous studies of CCF with optical trapping, we did not report any change in the trap stiffness along the protocol. The index of refraction scales as the density, i.e. $\delta n/n \propto \delta \rho/\rho$. However, the measure based on light scattering are based either on the gradient or on the Laplacian of the index of refraction field. Therefore the fluctuations are strongly enhanced by the derivatives.

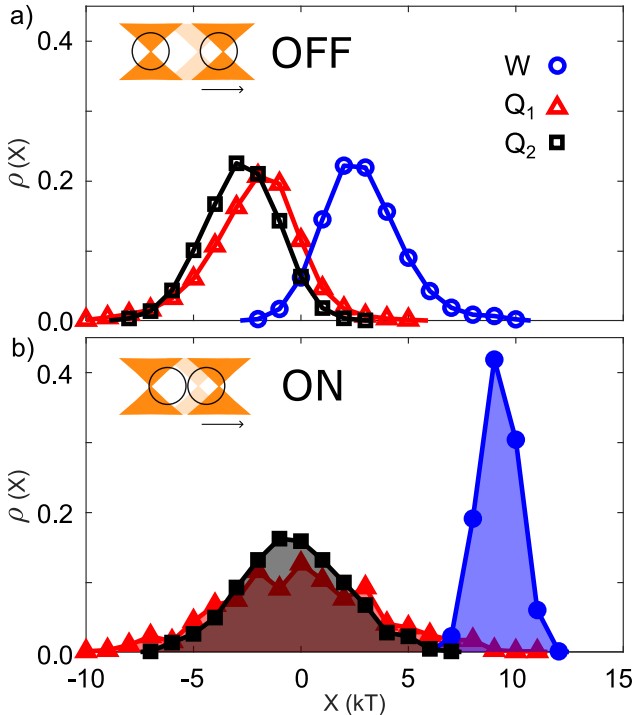

Figure 5: Probability density function of the energetics ($W$ blue circles, $Q_1$ red triangles, and $Q_2$ black squares) discriminating between a) ON events and b) OFF events (colored areas with full symbols and non-shadowed areas with empty symbols respectively) during the forward process at $P = 150$ mW. The decoupling of the energetics between the two different possibilities shows a higher dissipation during OFF events inferred from the non-zero mean distribution of both released heats (black and red open symbols with non-colored curve) compared with the zero mean heat distributions of the ON events (black and red full symbols with colored area). The difference between the work distributions of the ON and OFF events comes from the storage of free energy in the case of ON (both particles within the critical Casimir well).

Fluctuation theorems are powerful theoretical tools that generalize the second law of thermodynamics allowing us to connect out-of-equilibrium measures with equilibrium magnitudes like free energy or entropy changes. In particular, Crooks' theorem is very visual for discriminating between pure-dissipative and non-pure dissipative processes. Being $kT \log\left[\rho_F(W)/\rho_B(-W)\right] = W - \Delta F$, on the one hand, if $\Delta F = 0$ (all the injected work is dissipated as released heat) both distribution cross at zero. On the other hand, if there is a storing or release of energy in the system, i.e. $\Delta F \neq 0$, the pdfs will cross at $W = \Delta F$. Since in our system, the initial and final states are multiple-defined, as shown in previous sections, we build the pdf corresponding to each event (ON/OFF) and we can extract information. Namely, Fig. 4 shows multimodal distributions for $\rho_F(W)$ and $\rho_B(-W)$. However, we can easily notice that the distributions are not symmetric. This is due to the different contributions of the ON and OFF events. While the ON events activate the critical well, increasing $\Delta F$ along the forward protocol or decreasing it along the backward, the OFF events only dissipate the injected work as released heat. This is the reason why the forward and backward distribution of the ON events are almost identical (a very small amount of energy is dissipated) while the OFF distributions crosses each other at $W = 0kT$, see Fig. 5. However, we expect that an increase in the resolution for a given $\Delta T$ will reveal differences between $F$ and $B$ also at the OFF events. The energy dissipation is easier to visualize in Fig. 5b), where we show the heat distribution

for both particles as well as the work injected by selecting only the desired realizations. We observe how in the case of ON events the heat distribution is centered almost at zero, for both particles, while the distribution does not peak at zero in the case of OFF events. Indeed, the values of the mean work and mean heat shown in Fig. 3c) mainly evolve due to the changes in the probability of choosing between the different events. We want to point out how the distributions of work forward and backward overlap each other in the case of the ON events. We interpret it as the fact that the critical Casimir force is always in equilibrium for those displacements due to its fast relaxation time.

## 5  Conclusions

In this article, we have used light-induced heating to control the bath-induced forces between two optically trapped colloidal particles. The heating laser beam is focused by the upper objective irradiating the sample with a cylindrical shape that allows us to control the temperature in a range of 1K below the demixing point ($\approx 30.1^{\text{o}}$C) with a precision of $\pm 10$mK. The temperature is smoothly changed during hours with a symmetric protocol for ensuring quasistatic warming. We do not observe any hysteresis in the sample. Since the temperature calibration is done via the correlation length of the fluid obtained by non-linear fits to the total energy landscape, the precision of the method is highly affected by the duration of the measurement. To have an extra observable with the same optical setup, we study how the luminosity of the pixels has a clear dependence on the critical distance. However, since the scattering geometry is highly non-trivial and out of the interest of this article, the results remain open to link them with a quantitative value of the correlation length.

The temperature modulation via laser-induced heating is applied in a two-particle toy model whose flux of energy is changed as a function of the local temperature. The temperature control is stable enough to study the stochastic energetics of the two-particle system from a statistical point of view and to use the fluctuation theorems in the interpretation of the dissipation sources. From the use of the Crooks' theorem, we interpret that the critical Casimir force is always in equilibrium since the distributions of the work in the forward and backward process for the ON events overlaps each other. However, the light-induced heating technique paves the way for time-driven protocols in the local temperature for future experiments in the out-of-equilibrium performance of critical Casimir forces such as its equilibration or its influence in the rheology of the sample.

As future applications, it is pertinent to highlight two potential approaches for harnessing useful work from the depicted scheme. Firstly, the system's dual-equilibrium configuration exhibits a notable resemblance to the colloidal Szilard engine [13]. By effectively manipulating the probabilities associated with each equilibrium position, one can intentionally change the system's ergodicity, thereby facilitating the extraction of valuable work within the framework of thermodynamics of information. Secondly, a more advantageous avenue emerges. By carefully adjusting the temperature differentials during the forward and reverse processes, it becomes possible to readily access and extract the stored energy in the form of free energy. This rudimentary model aligns with the foundational principles that underlie the impact of critical interactions on biological systems [14]. .

We are just scratching the possibilities of fluctuation-induced forces in the performance of nanomachines or their use in nanothermometry. Indeed, although the range would be limited, critical interactions may be used in specific experiments whose temperature dependence is large in a narrow temperature range. In line with the final sentence of [15], "the consciousness of the environment as a part of the whole system is important not only in the ecology but also at the micron or nanoscale physics", we expect these results pave the way to future experiments where the thermal bath is not just a passive actor but a tunable agent during a physical process.

## Acknowledgments

**Funding information** IAM acknowledges funding from the Spanish Government through grants Contract (FIS2017-83709-R) and Juan de la Cierva program as well as the CNRS visiting researcher program and the MSCA-IF NEQLIQ - 101030465. AP and SC acknowledge funding from ANR18-CE30-0013. This work was partially financed by ERC-OUTEFLUCOP.

## A  Methods

*Experimental methods.* Our experiments are done in a low critical temperature micelle-solvent solution, $C_{12}E_5$ in Milli-Q water at 1.2% mass concentration. This mixture has a correlation length of $\xi_0 \approx 1.4$nm and a critical temperature $T_C \approx (30.5 \pm 0.1)°$C [16, 17]. A few microspheres (Fluka silica, $R = (2.50 \pm 0.35)\mu$m) per milliliter are added to the final mixture in a low concentration to allow long-term measurements without intrusive particles. The mixture is injected into a custom-made cell $40\mu$m thick and mechanically sealed to avoid contamination. This chamber is made by sandwiching a parafilm mask between a microscope slide and a sapphire optical window. The chamber thickness is reduced at minima to avoid thermophoretic or convective effects. Within the fluid cell, the two optical traps are created by a near-infrared laser beam (LaserQuantum, $\lambda = 1064$nm ) which is focused thanks to a high NA immersion oil objective (Leica ×63, NA=1.4). The dynamic optical trap is based on the control of the laser beam by an acousto-optical deflector (AA optoelectronics) which allows us to create two different traps using the time-sharing regime at 10 kHz. The two optical traps are kept $15\mu$m from the cell bottom slide. The bead's images are acquired by a high-speed camera (Mikrotron MC1310) and their positions are tracked in real-time by suitable software. The tracking resolution is $\pm 5$ nm with a conversion rate of $S = 105.4$nm/px. The acquisition frequency is fixed at 400 frames per second for all experiments. Our particle tracking is restricted to the XY plane while we only analyse the trajectory in the x-axis. We neglect the cross-coupling between the two axes (x and y) since this perturbation is second order in the Rotne-Prager diffusion tensor. The optical traps are calibrated using standard methods such as power spectrum density. From the time series, we obtain the total energy potential by the Boltzmann relation $\rho(r) \propto \exp(-U(r)/kT)$. The total potential can be split into its different components: electrostatical $U_{el}$, Casimir $U_{CCF}$ and trapping $U_{OT}$. The Critical Casimir contribution, $U_{CCF}$ is fitted assuming the Derjaguin approximation (for $d/R \ll 1$). The correlation length is obtained from the non-linear fit of $U_{cas}$ at different values of the irradiating power $P_h$ and hence $\Delta T = T_c - T$.

The ground temperature ($T_g$) is controlled by a double feedback system one on the objective and one inside the cell. Temperature is registered with two independent sensors (Pt 1000$\Omega$) and sent to a programmable temperature controller (PID Stanford research instruments). The objective and the cell are heated with heater mats (*Minco* 40 $\Omega$ and 80 $\Omega$ respectively). The whole system is thermally isolated from the environment to reduce the effect of environmental perturbations both on the position of the particles and on the temperature. Once the bulk is thermalized, $T_g = 28.00°$C the sample is irradiated using an infrared *heating* laser (Thorlabs, $\lambda = 1550$nm) focused by the upper objective (Leica, NA 0.53). The heating laser beam has a width of $20\mu$m and a depth of field of $400 \ \mu$m that is much larger than the chamber thickness ($40 \ \mu m$). Therefore, a cylindrical shape can be assumed in the description of the irradiating heating beam.

*Stochastic thermodynamics.* Work ($W$) and heat ($Q$) as the exchange of energy of the colloidal system with the external agent (change in the movable trap position $x_T$) and the thermal bath respectively. Work and heat are defined as $\delta W_i = \kappa(x_i - x_i^T) \circ \mathrm{d}x_i^T$ and

$\delta Q_i = -\kappa(x_i - x_i^T) \circ dx_i$, where $i = 1, 2$ are the two particles and $\circ$ stands for Stratonovich integration. From this definition, the work is always injected in particle 2, since $x_1^T$ is static. The events are discriminated between ON and OFF by comparing the mean distance during a single protocol with a threshold between the colloidal surfaces: if the mean relative distance is always below $d < 400$ nm along a single protocol, we consider it as an ON event. Ensemble average $\langle X \rangle$ of any quantity $X$ over $N$ process of $M$ points each is defined as $\langle X(t_j) \rangle = \sum_{i=1}^{N} \sum_{k=1}^{j} \delta X_i(t_k)$.

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
