# Peer review of "Laser-induced heating for the experimental study of critical Casimir forces with optical trapping"

_SciPost Physics, doi:SciPost Phys. 15, 247 (2023)_

## Round 1 · Author Response

Dear Editor, please, find below our discussion with the referees. First of all, we want to acknowledge both referees for their kind and swift comments. The manuscript has been changed according to their suggestion.

  1. Variations of the temperature will also change the order parameter profile around the particle. Due to the difference in dielectric properties of the two phases (which is exploited by the authors for the measurement of temperature) this may also influence the optical trapping. Can the authors comment on that?

We acknowledge the referee for pointing out it. Indeed, the trap stiffness is a function of the ratio between the refractive index of the media and the refractive index of the colloidal particle. The refractive index scales with the media density, which changes with a small universal exponent, hence its influence can be neglected. Nevertheless, the measurement relying on light scattering hinges on either the gradient or the Laplacian of the refractive index field. Consequently, the variations are significantly amplified by these derivatives.

  1. Did the authors also consider vertical changes in the particle position during their protocol?

We disregarded the second-order effects of the hydrodynamic tensor, particularly the orthogonal coupling. In future studies, it will be important to consider the cross-motion to assess potential alterations in the drag term, specifically for microrheological investigations of the sample.

  1. It would be helpful, the authors could discuss the differences between their engine and a colloidal Carnot or Stirling engine. My impression is, that apart from the technical realization, there are large similarities. In both cases the available phase volume of the colloid is periodically changed which leads to the extraction of work.

Our system differs significantly from conventional Carnot or Stirling cycles due to the absence of two thermal reservoirs from which useful work is extracted. Our primary objective is to scrutinize energy current fluctuations when the driving force originates from a thermal bath. Interestingly, our two-state system can be reinterpreted as a Szilard engine if we engineer the balance between its two potential wells. This augmentation introduces the intriguing dimension of the thermodynamics of information. Furthermore, leveraging the unique properties of critical Casimir forces, we can envision an information-emerging system where the role of information, such as a sequence, can be toggled on and off with minor temperature adjustments.

---

## Round 1 · List of Changes

Requested changes

1. Variations of the temperature will also change the order parameter profile around the particle. Due to the difference in dielectric properties of the two phases (which is exploited by the authors for the measurement of temperature) this may also influence the optical trapping. Can the authors comment on that?

We added the following sentence:

As in previous studies of CCF with optical trapping, we did not report any change in the trap stiffness along the protocol. The index of refraction scales as the density, i.e. Delta n/n \propto \Delta \rho/\rho. However, the measures based on light scattering are based either on the gradient or on the Laplacian of the index of refraction field. Therefore the fluctuations are strongly enhanced by the derivatives.

2. Did the authors also consider vertical changes in the particle position during their protocol?

We added the following sentence:

Our particle tracking is restricted to the XY plane while we only analyse the trajectory in the x-axis. We neglect the cross coupling between the two axis (x and y) since this perturbation is second order in the Rotne-Prager diffusion tensor.

3. It would be helpful, the authors could discuss the differences between their engine and a colloidal Carnot or Stirling engine. My impression is, that apart from the technical realization, there are large similarities. In both cases the available phase volume of the colloid is periodically changed which leads to the extraction of work.

We added the following paragraph:

As future applications, it is pertinent to highlight two potential approaches for harnessing useful work from the depicted schematic. Firstly, the system's dual-equilibrium configuration exhibits a notable resemblance to the colloidal Szilard engine. By effectively manipulating the probabilities associated with each equilibrium position, one can intentionally disrupt the system's ergodic behavior, thereby facilitating the extraction of valuable work within the framework of thermodynamics of information. Secondly, a more advantageous avenue emerges. By carefully adjusting the temperature differentials during the forward and reverse processes, it becomes possible to readily access and extract the stored energy in the form of free energy. This rudimentary model aligns with the foundational principles that underlie the impact of critical interactions on biological systems.

---

## Editorial Decision

published